# Chymotrypsin is a molecular target of insect resistance of three corn varieties against the Asian corn borer, *Ostrinia furnacalis*

**Eun Young Kim[1,2], Jin Kyo Jung[2], I. Hyeon Kim[2], Yonggyun Kim[1]***

**1** Department of Plant Medicals, Andong National University, Andong, Korea, **2** Division of Crop Cultivation and Environment Research, Department of Central Area Crop Science, National Institute of Crop Science, Rural Development Administration, Suwon, Korea

* hosanna@anu.ac.kr

**Data Availability Statement:** All relevant data are within the paper and its Supporting Information files.

**Funding:** This research was supported by a "Cooperatibe Research Program for Agriculture

## Abstract

The Asian corn borer, *Ostrinia furnacalis*, is a serious insect pest that can infest corn leaves and stems. Due to its internal feeding behavior, its larvae are not exposed to insecticides that are usually sprayed for pest control. To minimize crop damage caused by *O. furnacalis*, improving insect resistance trait of corn has been considered as an optimal control tactic. This study screened 27 corn varieties for their insect resistance trait and selected three varieties of Ilmichal (IM), P3394, and Kwangpyeongok (KP) that showed insect resistance trait. *O. furnacalis* larvae did not show any significant difference in preference between these three insect-resistant corn varieties and a control susceptible variety. However, these resistant varieties after ingestion significantly interfered with larval development of *O. furnacalis*. This suggests that the insect resistance trait is induced by antibiosis, but not by antixenosis. Indeed, larvae fed with these varieties suffered from low chymotrypsin (CHY) activities in the midgut juice. To determine the target CHY inhibited by resistant corn varieties, a total of nine CHY genes (*Of-CHY1~Of-CHY9*) were predicted from the transcriptome of *O. furnacalis*. Six genes (*Of-CHY1~Of-CHY6*) were expressed in all developmental stages and tissues. Especially, *Of-CHY3* was highly expressed in the midgut of *O. furnacalis* larvae. RNA interference (RNAi) using double-stranded RNA (dsRNA) specific to *Of-CHY3* (2 µg of dsRNA injected to each L5 larva) resulted in significant reduction of *Of-CHY3* expression level at 24 h post-treatment. Feeding L3 larvae with this dsRNA also significantly suppressed the expression level of *Of-CHY3* and reduced its enzyme activity at 24 h post-treatment. A recombinant *Escherichia coli* expressing dsRNA specific to *Of-CHY3* was constructed using L4440 vector. Feeding such recombinant bacteria suppressed the expression level of *Of-CHY3* and prevented larval development of *O. furnacalis*. These results suggest that the three resistant varieties can produce a resistance factor(s) to inhibit the CHY activity of *O. furnacalis* and suppress larval growth. This study suggests that CHY might be an inhibition target in *O. furnacalis* for breeding insect-resistant corns.

Science & Technology Development (Project No.
PJ01503802)" funded by Rural Development
Administration, Republic of Korea to EYK. This
work was also supported by a Research Grant of
Andong National University, Republic of Korea to
YK.

**Competing interests:** The authors have declared
that no competing interests exist.

## 1. Introduction

The Asian corn borer, *Ostrinia furnacalis*, has spread from East/Southeast Asia to the western
Pacific islands, where it is the most serious insect pest infesting corns. With insecticidal resistance and internal feeding behavior, chemical control for this insect pest is inefficient [1].
Environmentally favorable and sustainable cropping needs non-chemical control tactics such
as breeding insect-resistant corns against *O. furnacalis*.

Upon herbivorous insect attacks, host plants have evolved diverse defensive strategies
[2]. These strategies can be divided into resistance and tolerance [3]. Resistance includes
chemical and physical deterrence of insect settling and feeding (= antixenosis) and reduced
plant palatability (= antibiosis) by expressing toxic compounds to impair insect gut function [4, 5]. Tolerance is a plant trait that can maintain or promote plant fitness following
insect damage [6]. These defense traits can be constitutive or inducible. Constitutive
defenses are the basal expression of physical and chemical defensive traits in the absence of
herbivores, whereas induced defenses are mounted only after herbivorous insect attack [7].
Indeed, corn plants are known to induce insect resistance traits upon feeding damage by *O.
furnacalis*, in which defence-related gene expression is induced at 2 h after feeding along
with phytohormones and subsequent volatiles to attract *Macrocentrus cingulum*, a parasitic
wasp [8].

When damaged by herbivorous insects, some corn varieties can produce defensive
enzymes, phenols, proteinase inhibitors, and benzoxazinoids against corn borers. For example,
2,4-dihydroxy-7-methoxy-1,4-benzoxazine-3-one, commonly known as DIMBOA, has been
reported in several gramineous species, including maize, wheat, and rye [9]. In response to
infestation by European corn borer, *Ostrinia nubilalis*, levels of glucose derivative (HDIM-
BOA-Glc) of DIMBOA are increased in corn stems [10]. These DIMBOA compounds can
inhibit insect growth and development by inhibiting digestive proteases (such as trypsin and
chymotrypsin) and detoxification enzymes [11–13]. Defense-related enzymes in plants such as
polyphenol oxidase, superoxide dismutase, catalase, and peroxidase catalyze the production of
toxic secondary metabolites against hervibores [14–16]. Insects have also developed specific
and efficient adaptation mechanisms. For example, *O. furnacalis* has developed an efficient
degradation using UDP-glucosyltransferase or other UDP-glucose-dependent enzymes [17],
suggesting a coevolution between plants and herbivorous insects.

Serine proteases such as trypsin and chymotrypsin (CHY) that function in the gut are the
main proteolytic digestive enzymes in lepidopteran insects [18]. Catalytic triad of His-Ser-Asp
is conserved at their active sites. Sessile peptide bond is cleaved by a nucleophilic attack of an
ionized serine. The resulting small protein fragments are then digested by amino- or carboxy-
peptidases to generate free amino acids for insect growth and development [19]. Herbivorous
insects suffer from the lack of protein sources from plant hosts. They exhibit a voracious feeding behavior to meet their protein requirement [20]. Thus, any inhibition of protein digestion
can critically intimidate the survival of insects.

To develop an efficient control technique against *O. furnacalis* by altering midgut digestion,
this study screened insect-resistant corn strains against *O. furnacalis*. These selected resistant
corn varieties inhibited CHY activity in *O. furnacalis* and resulted in their developmental
retardation. To test a hypothesis that the inhibitory activity of insect-resistant corn strains
against CHY activity in *O. furnacalis* could lead to their developmental retardation, this study
used RNA interference (RNAi) with a loss-of-function approach. A specific RNAi against *O.
furnacalis* CHY expression was effective in giving adverse effects on the larval development. A
recombinant bacterium expressing this specific dsRNA further supported the antibiosis
hypothesis of these insect-resistant corns for inhibiting CHY activity in the host.

## 2. Materials and methods

### 2.1. Insect rearing

Larvae of *O. furnacalis* were reared on an artificial diet [21] under laboratory conditions (25˚C, 15 h photophase, and 60 ± 5% relative humidity). Under these conditions, *O. furnacalis* underwent five larval instars (L1-L5). Adults were supplied with 10% sucrose solution.

### 2.2. Plant culture

Each of four corn varieties (Ilmichal (IM), Kwangpyeongok (KP), P3394, and Gangwonchal 60 (GC60)) was sowed in a pot (170 cm diameter, 147 cm height) (Green Supplies, Goyang, Korea) containing bed soil (Barokea, Seoulbio, Seoul, Korea). Three corn varieties were culti-vated under greenhouse conditions (35 ± 10˚C, 15 h photophase). All experiments used 4th to 7th leaves except cotyledon at V7 corn stage (defined as seven leaves with collar visible). The first leaf with the rounded tip was included [22, 23].

### 2.3. Chemicals

SAAPFpNA(N-succinyl-alanine-alanine-proline-phenylalanine-*p*-nitroanilide), chymostatin, tosyl phenylalanyl chloromethyl ketone (TPCK), and tosyl-L-lysyl-chloromethane hydrochlo-ride (TLCK) were purchased from Sigma-Aldrich Korea (Seoul, Korea). Cathepsin III inhibi-tor (CATH) was purchased from Calbiochem (San Diego, CA, USA). Substrate and inhibitors were all dissolved in dimethyl sulfoxide (DMSO). Phosphate-buffered saline (PBS) was pre-pared with 100 mM sodium phosphate containing 0.7% NaCl and adjusted to pH 7.4.

### 2.4. Primary screening of insect resistance for different corn varieties

Two or three seeds of each corn variety were sowed in a hole with a spacing of $25 \times 70$ cm. The soil was treated with a fertilizer (20 (nitrogen)–15 (phosphorus)–15 (potassium) kg/10 a) by mixing N-SuperAlal-e (Namhea chemical corp., Yeosu, Korea), P-Younggwarin (KG chemical, Ulsan, Korea), and K-Potassium chloride (Farmhannong, Seoul, Korea). Briefly, newly hatched larvae of *O. furnacails* were applied at a density of about 150–200 larvae to individual corn plant at V6 corn stage by the bazooka method [24]. At 40 days after the application, insect resistance of corn varieties was scored according to the resistance method [25]. Briefly, the resistance intensity was scored from a scale of 1 to 9 according to the number of holes and frass in corn stalk. The scoring system had '1–4' grades for shoot holes, and '5–9' grades for elongated lesions. The following grading system was used for the resistance scoring induced by corn damage.

Grade 1: No injury to sheat-collar tissue, no visible holes in stalk, no frass.

Grade 2: Up to 5% sheath-collar damage, very little or no holes in stalk visible and frass.

Grade 3: Up to 10% sheath-collar damage, very few holes in stalk and some frass.

Grade 4: Up to 20% sheath-collar damage, intermediate number of visible holes in stalk and frass.

Grade 5: Up to 30% sheath-collar damage, intermediate number of visible holes in stalk and frass.

Grade 6: Up to 40% sheath-collar damage, intermediate number of visible holes in stalk and frass.

Grade 7: Up to 50% sheath-collar damage, several visible holes in stalk, a lot of frass.

Grade 8: Up to 75% sheath-collar damage, several visible holes in stalk, a lot of frass.

Grade 9: Up to 100% sheath-collar damage, a lot of visible holes in stalk, a lot of frass.

Less than grade '3' indicated high resistance ('HR'). Grades of '3–5' indicated resistance ('R'). Gades of '5–7' indicate intermediate resistance ('IR'). Grades greater than '7' indicated susceptible ('S') varieties.

## 2.5. Feeding preference test

Four corn varieties ('IM', 'KP', P3394, and 'GC60') were used to compare the feeding preference of *O. furnacalis*. The susceptible variety ('GC60') was used as a reference. Young L3 larvae (within 9 h after molting) were starved for 6 h just before test. Leaves of four corn varieties were cut at the proximal area containing the stalk, which was wrapped with wet cotton to prevent desiccation. Four leaves representing each of four varieties was placed in a Petri dish (130 mm diameter, 25 mm height) (Green Supplies, Goyang, Korea) in a quarter part of the Petri dish, respectively. In a treatment (= Petri dish), 10 starved larvae were placed in the center of the dish. Each treatment was replicated three times. After 24 h, the number of the larvae reaching a specific corn variety was counted. Feeding amount of each leaf was measured by weight loss for 2 days after initiation of the trial. The feeding amount was corrected by the weight loss of the same size of leaf due to desiccation under the same environmental conditions during the assay.

## 2.6. Determination of *O. furnacalis* growth parameters

Newly hatched larvae were fed with different corn leaves as described above. Larval growth parameters included larval period, frequency of supernumery molts (= L6 or L7 larvae), pupal weight, and adult emergence. Artificial diet was used as control under the same rearing conditions. Each treatment was replicated three times. Each replication used 30 larvae.

To further assess the adverse effects of corn leaves on *O. furnacalis* development, corn leaves were freeze-dried by Biobase (FD8508, Ilshin, Dongducheon, Korea) and pulverized. The freeze-dried powder was then added to the artificial diet (S1 Table). These modified artificial diets were fed to newly hatched larvae to determine growth parameters as described above.

## 2.7. Influence of protease inhibitors in *O. furnacalis*

After starving for 6 h, third instar larvae were treated with artificial diet treated with different protease inhibitors: chymostatin specific to CHY, tosyl-phenylalanyl-chloromethyl ketone (TPCK) specific to CHY, tosyl-L-lysyl-chloromethane hydrochloride (TLCK) specific to trypsin, and cathepsin III inhibitor (CATH) specific to catheptin. All inhibitors were dissolved in 10% dimethyl sulfoxide (DMSO) to prepare stock solution at 10, 100, and 1,000 ppm. L3 larvae were fed artificial diet (0.2 × 2.0 × 0.1 cm) overlaid with 30 μL inhibitor solution for 4 days. Treated larvae were then supplied with fresh diet for 4 days. Survival rates were determined at 8 days after treatment initiation. Larves in all treatment groups were reared at laboratory conditions (25°C, 15 h light photophase, and 60 ± 5% relative humidity). Each treatment was replicated three times. For each replication, 10 larvae were used. As a control, 10% DMSO (30 μL) without any inhibitor was used to treat the diet.

## 2.8. CHY enzyme assay of *O. furnacalis*

Gut juice of *O. furnacalis* was collected from three L3 or L5 larvae, respectively. All collected larvae were 2-day old. Extracted L3 or L5 larvae gut juice samples were homogenized in 300 μL

or 500 μL PBS, respectively, and centrifuged at $19,000 \times g$ for 10 min at 4°C. The supernatant was trasfered to a fresh tube as lumen sample. The amount of protein in the lumen sample was measured by Bradford assay using a protein assay reagent (Bio-Rad, Hercules, CA, USA). Standard curve was obtained using bovine serum albumin (BSA, Thermo Scientific, Waltham, MA, USA). CHY activity of *O. furnacalis* was measured using a synthetic peptide substrate SAAPFpNA (N-succinyl-alanine-alanine-proline-phenylalanin-p-nitroanilide). Briefly, 5 μL of 1 mM substrate was added to 10 μL enzyme extract. Then 85 μL of 50 mM Tris-Hcl buffer (pH 7.5) was added. The mixture was incubated at room temperature for 30 min. The product was measured on a 96-well microplate reader (Molecular devices, San Jose, CA, USA) at wavelength of 410 nm [26]. CHY activity unit (U) was expressed as a change in absorbance per minute per microgram protein. The substrate molar extinction coefficient was $8,800 \text{ M}^{-1}\text{cm}^{-1}$ at 410 nm.

### 2.9. Bioinformatics to predict CHY (*Of-CHY*) genes

Nine CHY-like genes (*Of-CHY1 ~ Of-CHY9*) of *O. furnacalis* were obtained from NCBI GenBank (www.ncbi.nih.nlm.gov). Predicted amino acid sequences were aligned using Clustal W. Phylogenetic tree was constructed with the Neigbor-joining method. To obtain the parsimony of the branching, bootstrap values were obtained with 1,000 replications (Mega X, www. megasoftware.net). The signal peptide in predicted *Of-CHY* genes was predicted using the SignalP 5.0 server (http://www.cbs.dtu.dk/services/SignalP/index.php). The protein of *Of-CHY* domain was predicted with the PROSITE server (https://prosite.expasy.org).

### 2.10. Tissue preparation of *O. furnacalis*

Different tissues were prepared from 2-day old L5 larvae. After longitudinally cutting the ventral integument, the whole intestine was isolated. To obtain the midgut, the foregut (a relatively slender area near the mouth) and the hindgut (posterior to Malpighian tubule) were removed. Fat bodies were then collected by scratching the remaining body part of the larvae with blunt end forceps. The residue after collecting fat body was collected as the epidermis. For each type of tissues, three larvae were used. Each treatment was replicated three times.

### 2.11. RNA extraction and cDNA preparation

Total RNA was extracted from the whole body of *O. furnacalis* at different developmental stages (100 eggs, 20 L1 larvae, ten L2 larvae, five L3 larvae, three L4 larvae, one L5 larva, one pupa, or one adult) using Trizol solution (ProGEN, Jeonju, Korea) according to the manufacturer's protocol. RNA samples were stored at -70°C before use. cDNA synthesis was performed using RT-premix (Intron Biotechnology, Seongnam, Korea) and 300 ng total RNA sample following the manufacturer's instructions.

### 2.12. RT-qPCR analysis of *Of-CHY* gene expression

PCR amplification was performed using cDNA as template and gene-specific primers (S2 Table). PCR was performed under the following conditions. Each PCR reaction (20 μL) contained 10 μL 2x PCR premix (GeneAll, Seoul, Korea), 1 μL of cDNA template, 1 μL of forward and reverse primers (10 pmol/μL) each, and 7 μL distilled and deionized water (ddH$_2$O). PCR reaction began with an initial denaturation step at 95°C for 5 min, followed by 35 cycles of denaturation at 95°C for 1 min, annealing at 54 or 50°C for 20 sec, and extension at 72°C for 20 sec, and a final extension step at 72°C for 5 min.

Quantitative PCR (RT-qPCR) was performed using SYBR Green Supermix (Bio-Rad Laboratories, Hercules, CA, USA) and a CFX PCR machine (CFX96, Hercules, CA, USA) according to the manufacturer's instruction. The reaction mixture contained 10 μL SYBR Green Supermix, 100 ng of cDNA template, 1 μL of forward and reverse primers (10 pmol/μL) each, and ddH$_2$O to have a final volume of 20 μL. After an initial denaturation at 95˚C for 5 min, cDNA was amplified with 40 cycles of denaturation at 95˚C for 1 min, annealing at 54 or 50˚C for 20 sec, and extension at 72˚C for 20 sec, followed by a final extension step at 72˚C for 5 min. Fluorescence values were measured and amplification plots were generated in real time. Melting curves of PCR products were analyzed to confirm single products. Quantitative analysis of gene expression was performed using the $2^{-\Delta\Delta CT}$ method [27]. Expression levels of a ribosomal gene, *RL32*, in different samples were measured to confirm the cDNA preparation and normalize target gene expression levels [28]. Each treatment was replicated with three independent biological samples.

## 2.13. RNA interference (RNAi) of *Of-CHY3* gene expression

Double-stranded RNA (dsRNA) was generated *in vitro* using Megascript RNAi Kit (Invitrogen, MA, USA) according to the manufacturer's instruction. Using cDNA mentioned above, about 273 bp-long PCR product of *Of-CHY3* was amplified with gene-specific primers containing T7 promotor sequence at the 5' end (*dsCHY3-T7* in S2 Table). The resulting PCR product was used as template for *in vitro* transcription using T7 RNA polymerase provide by the kit. The synthesized dsRNA was purified with absolute ethanol and subjected to 1.5% agarose gel electrophoresis to confirm the single product. Control dsRNA ('*dsCON*') was also generated using *EGFP* (enhanced green fluorescence protein) gene as a template (included in the kit).

For dsRNA injection, 3 μL of *dsOfCHY3* was injected to each L5 larva through abdominal proleg with a microsyringe (SGE, Sydney, Australia). For dsRNA oral delivery, 6 h-starved L3 larvae were treated with an artificial diet (0.2 × 2.0 × 0.1 cm) overlaid with *dsOfCHY3* solution. After a complete dsRNA-treated diet consumption (within 24 h), the consumed dsRNA concentration was calculated by dividing by the number of treated larvae per diet. For each treatment, ten larvae were used. Each treatment was replicated three times.

## 2.14. Constructing a recombinant *Escherichia coli* expressing dsRNA specific to *Of-CHY3* and its overexpression

A target fragment of *Of-CHY3* (273 bp) was inserted to pCR2.1-TOPO vector (Invitrogen, Waltham, MA, USA) by TA cloning technique as described by the manufacturer. Using XhoI and HindIII restriction enzymes, the insert in the multiple cloning site of the vector was excised and recombined with an expression vector, L4440. The resulting recombinant vector (L4440-dsOfCHY3) was used to transform *E. coli* HT115 lacking RNase III by a heat shock method. Transformed bacteria were grown in Luria-Bertani (LB) medium containing 100 ppm ampicillin (AMP) at 37˚C with shaking (220 rpm) for 16 h. Bacteria at a growth phase detected by OD$_{600}$ = 0.5–0.7 were induced to overexpress dsRNA under T7 promoter by adding 1 mM (a final concentration) of isopropyl-*β*-D-1-thiolactopyranoside (IPTG). Bacteria were then incubated at 37˚C for 4 h. IPTG-induced cultures were pelleted by centrifugation at 8,000 rpm for 20 min and the pellet was re-suspended in PBS. Total bacterial RNA was extracted with Trizol solution (ProGEN, Jeonju, Korea). The presence of the synthesized dsRNA was confirmed by electrophoresis using 1.5% agarose gel with 1 x TAE buffer.

Quantification of dsRNA amounts in recombinant bacteria followed the method described previously [29]. A standard curve was generated with known amounts of purified dsRNA by

measuring gel band intensities with an image analyzer (ChemiDoc XRS, Bio-Rad, Hercules, CA, USA).

## 2.15. Feeding bioassay of an artificial diet treated with recombinant *E. coli* expressing dsRNA specific to *Of-CHY3*

The recombinant *E. coli* overexpressing dsRNA was harvested by centrifugation. The pellet was resuspended in 4 mL of PBS. The bacterial suspension was then sonicated using an ultra-sonicator (Sonics, Newtown, CT, USA) at 40% intensity with 10 cycles of 5 sec burst separated by a 5 sec gap. Bacterial viability after sonication treatment was assessed by plating 20 μL of each treated bacteria sample onto an LB+AMP plate. Bacteria treated by ultrasonication were used to prepare artificial diets.

To determine the insecticidal activity of recombinant bacteria, L3 larvae of *O. furnacalis* were used in a diet-dipping feeding assay. A piece of artificial diet ($2 \times 20 \times 1$ mm) was covered with $10^8$ cells in 20 μL. As control, HT115 with non-recombinant L4440 vector was used for the feeding assay. After complete consumption (within 24 h) of treated diet for 3 days, larvae were fed with fresh artificial diet for growth. After all bacterial cell consumption, consumed dsRNA and bacterial cell number were calculated by dividing the number of treated larvae per diet. Each treatment was replicated three times. For each replication, ten larvae were used. L1 larvae were reared in plastic cups (Frontier Agricultural Sciences, Newark, DE, USA). Other stage larvae were reared in Petri dishes (SPL, Pochoen, Korea). All developmental parameters such as larval and pupal ecdysis, pupal weight, and adult emergence were determined by daily monitoring.

## 2.16. Statistical analysis

For the standard curve, a linear regression was performed using Microsoft Excel program. All data were analyzed by one-way analysis of variance (ANOVA) and Tukey test ($\alpha = 0.05$) using PROC GLM of SAS program.

## 3. Results

### 3.1. Field scoring of different corn varieties for insect resistance

Different corn varieties were cultivated in fields and scored for their susceptibility to *O. furnacalis* infestation for five years (2017 ~ 2021). In this screening (Table 1), five sweet corn varieties (Danok 3, Godangok, Suwondan 85, Suwondan 87, and Suwondan 88), 12 waxy corn varieties (Gangwonchal 55, Gangwonchal 57, Gangwonchal 60, Gyeongbukchal 15, Gyeongbukchal 16, Gyeonggichal 4, Heukjinjuchal, Ilmichal, Mibeak 2, Suwonchal 84, Suwonchal 87, and Suwonchal 89), and 10 corn varieties for silage (Gangilok, Gangwon 46, Gangwon 52, Gangwon 55, Gangwon 58, Kwangpyeongok, Jangdaok, P3394, Suwon 215, and Suwon 226) were found to be relatively resistant to *O. furnacalis* infestation. Among these 27 corn varieties, three corn varieties (Ilmichal ('IM'), P3394 and Kwangpyeongok ('KP')) were selected to investigate their resistant characters because their resistant scores were relatively stable (less than 1.0 in standard deviation) during the five-year screening period. A susceptible corn variety (Gangwonchal 60 ('GC60') > 7.0 in decision value) was also selected for comparison.

### 3.2. Three resistant corn varieties show antibiosis to *O. furnacalis*

The three selected corn varieties along with the susceptible control were tested in host preference against *O. furnacalis*. Using choice test, the number of released L3 instar larvae was counted on each variety after 24 h with the susceptible variety ('GC60') as reference. There was

**Table 1. Screening 27 corn varieties against infestation by *O. furnacalis*.**

| Corn varieties | Yearly replication | | | | | Score[1] | Decision[2] |
|---|---|---|---|---|---|---|---|
| | 2017 | 2018 | 2019 | 2020 | 2021 | | |
| Danok 3. | 5.8 | 6.7 | 5.5 | 7.8 | 6.4 | 6.4 ± 0.9 | IR |
| Gangilok | 4.4 | 3.8 | 3.3 | 5.0 | 3.9 | 4.1 ± 0.7 | R |
| Gangwon 46 | 6.4 | 3.4 | 3.8 | - | - | 4.5 ± 1.6 | R |
| Gangwon 52 | - | 4.1 | 3.7 | 5.2 | - | 4.3 ± 0.8 | R |
| Gangwon 55 | - | - | 3.0 | 4.3 | 4.9 | 4.1 ± 1.0 | R |
| Gangwon 58 | - | - | 3.4 | 4.6 | 4.7 | 4.2 ± 0.7 | R |
| Gangwonchal 55 | - | 3.2 | 5.9 | 6.1 | - | 5.1 ± 1.6 | IR |
| Gangwonchal 57 | - | - | 4.2 | 4.9 | 6.1 | 5.1 ± 1.0 | IR |
| Gangwonchal 60 (GC60) | - | - | - | 8.6 | 7.5 | 8.1 ± 0.8 | S |
| Godangok | 4.0 | 3.6 | 4.0 | 6.4 | 4.4 | 4.5 ± 1.1 | R |
| Kwangpyeongok (KP) | 3.3 | 2.9 | 3.5 | 4.8 | 3.8 | 3.7 ± 0.7 | R |
| Gyeonbukchal 15 | - | 4.0 | 6.0 | 6.5 | - | 5.5 ± 1.3 | IR |
| Gyeonbukchal 16 | - | 4.8 | 5.9 | 7.2 | - | 6.0 ± 1.2 | IR |
| Gyeonggichal 4 | - | 3.3 | 5.8 | 6.5 | - | 5.2 ± 1.7 | IR |
| Hukjinjuchal | 4.3 | 2.5 | 5.3 | 5.2 | 4.3 | 4.3 ± 1.1 | R |
| Ilmichal (IM) | 5.9 | 3.8 | 6.1 | 5.8 | 5.5 | 5.4 ± 0.9 | IR |
| Jangdaok | 4.0 | 3.4 | 3.2 | 5.1 | 4.9 | 4.1 ± 0.8 | R |
| Mibaek 2. | 6.5 | 2.8 | 5.5 | 5.7 | 5.8 | 5.3 ± 1.4 | IR |
| P3394 | 5.0 | 4.8 | 4.3 | 5.8 | - | 5.0 ± 0.6 | IR |
| Suwon 215 | 6.0 | 3.9 | 5.2 | - | - | 5.0 ± 1.1 | IR |
| Suwon 226 | - | - | 2.9 | 4.5 | 5.6 | 4.3 ± 1.4 | R |
| Suwonchal 84 | 7.0 | 2.9 | 5.4 | 4.8 | - | 5.0 ± 1.7 | IR |
| Suwonchal 87 | - | - | 6.1 | 5.4 | 5.3 | 5.6 ± 0.4 | IR |
| Suwonchal 89 | - | - | 5.6 | 7.4 | 5.9 | 6.3 ± 1.0 | IR |
| Suwondan 85 | - | 3.5 | 3.6 | 5.4 | 3.6 | 4.0 ± 0.9 | R |
| Suwondan 87 | - | - | 6.9 | 7.9 | 3.4 | 6.0 ± 2.3 | IR |
| Suwondan 88 | - | - | 6.0 | 7.7 | 5.2 | 6.3 ± 1.3 | IR |

[1] Damage intensity ranged from 0 to 9.

[2] Resistance ('R'): Score ≤ 3; intermediate resistance ('IR'): 3 < score ≤ 5; susceptible ('S'): 5 < score ≤ 7.

Red- and blue-colored varieties were selected as resistant and susceptible varieties for subsequent study.

no marked difference in host preference between susceptible and resistant corn varieties (Fig 1A). Feeding activities were not also significantly different between susceptible and resistant corn varieties. These results suggest that *O. furnacalis* larvae do not discriminate the three resistant corn varieties in host preference.

Larvae of *O. furnacalis* were fed with three different corn variety leaves and their developmental parameters were then measured. Artificial diet ('AD') was used as a control diet and one susceptible variety was used for comparison with the three resistant corn varieties. After fed with AD, most larvae of *O. furnacalis* underwent five larval molts and pupated. However, almost half of larvae fed with the three corn varieties underwent supernumerary molts to six instar or additional larval stages (Fig 1B). The susceptible variety also induced an abnormal development. However, its effect was slightly weaker than the three resistant varieties. Furthermore, most larvae fed on the three corn varieties did not develop to pupal stage while more than 95% of larvae fed on artificial diet developed to pupae. These results suggest that the three resistant corn varieties have antibiosis factor(s) against *O. furnacalis*.

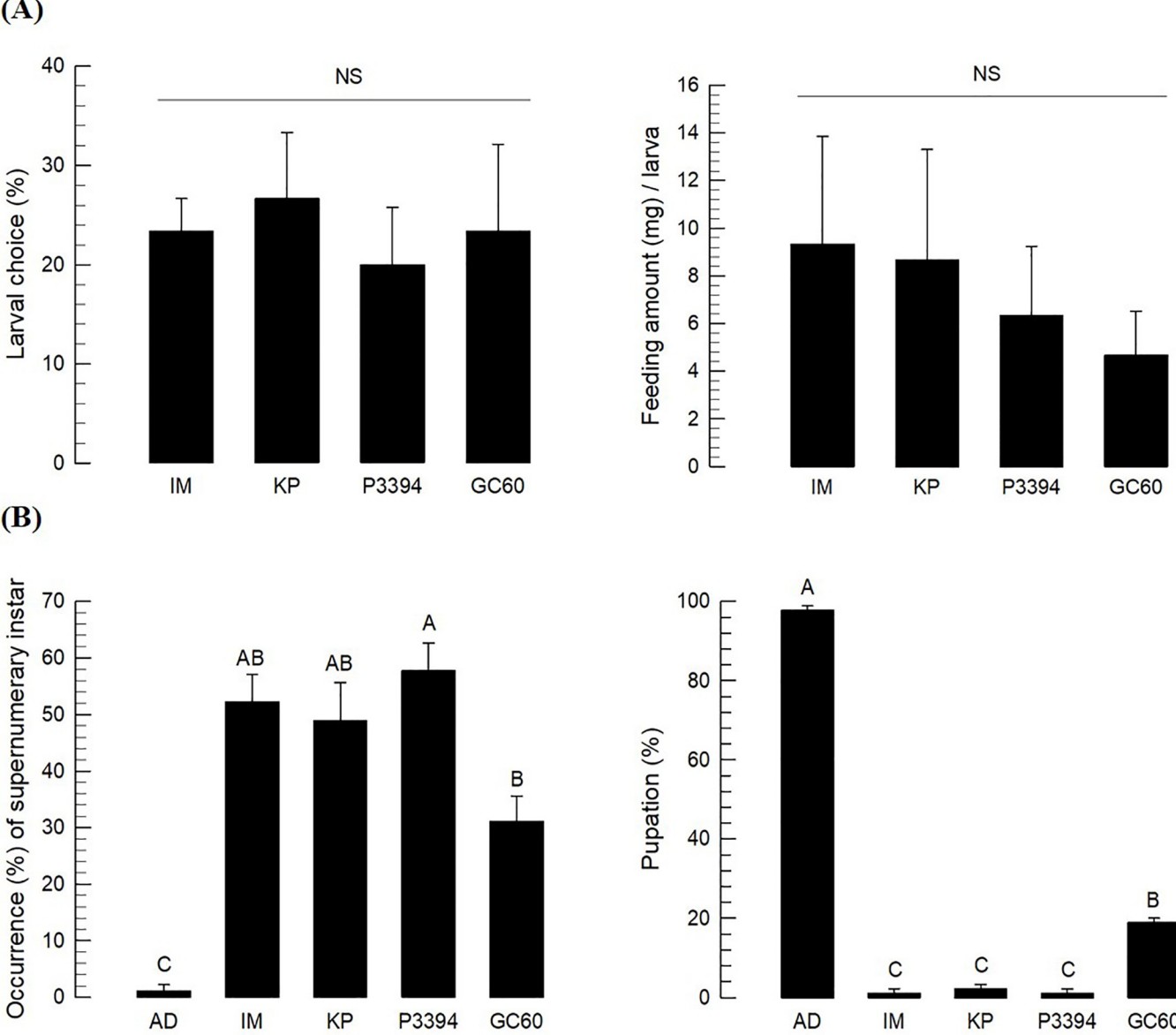

**Fig 1. Insect resistance assays of three resistant corn varieties (Ilmichal ('IM'), Kwangpyeongok ('KP') and P3394) and one susceptible (Gwangwonchal 60 ('GC60')) against *O. furnacalis*.** (A) Antixenosis assessment. L3 larvae were used in choice test. Each test was replicated three times. For each replication, 10 larvae were used. Left panel: Choice test of *O. furnacalis* among three corn varieties in Petri dish assay. Right panel: Variation in their feeding amount for 48 h. (B) Antibiosis assessment. 'AD' stands for artificial diet as a reference. Newly hatched larvae were reared with different diets until emergence. Each treatment was replicated three times. For each replication, 30 larvae were used. Different letters above standard error bars indicate significant difference among means at Type I error = 0.05 (Tukey test). 'NS' stands for no significant difference.

To test the hypothesis that the three corn varieties could produce antibiosis factor(s) against *O. furnacalis*, different amounts of a resistant variety ('KP') leaf powder were added to the artificial diet instead of wheat germ. These modified artificial diets showed significant adverse effects on larval development of *O. furnacalis* (Fig 2). The larval period was prolonged by the corn leaf powder in a dose-dependent manner (Fig 2A). Supernumery instars such as 6th or 7th instars were generated after treatment with 80% corn leaf powder. The adverse effect of a resistant variety ('KP') on the larval development of *O. furnacalis* was compared with those of other

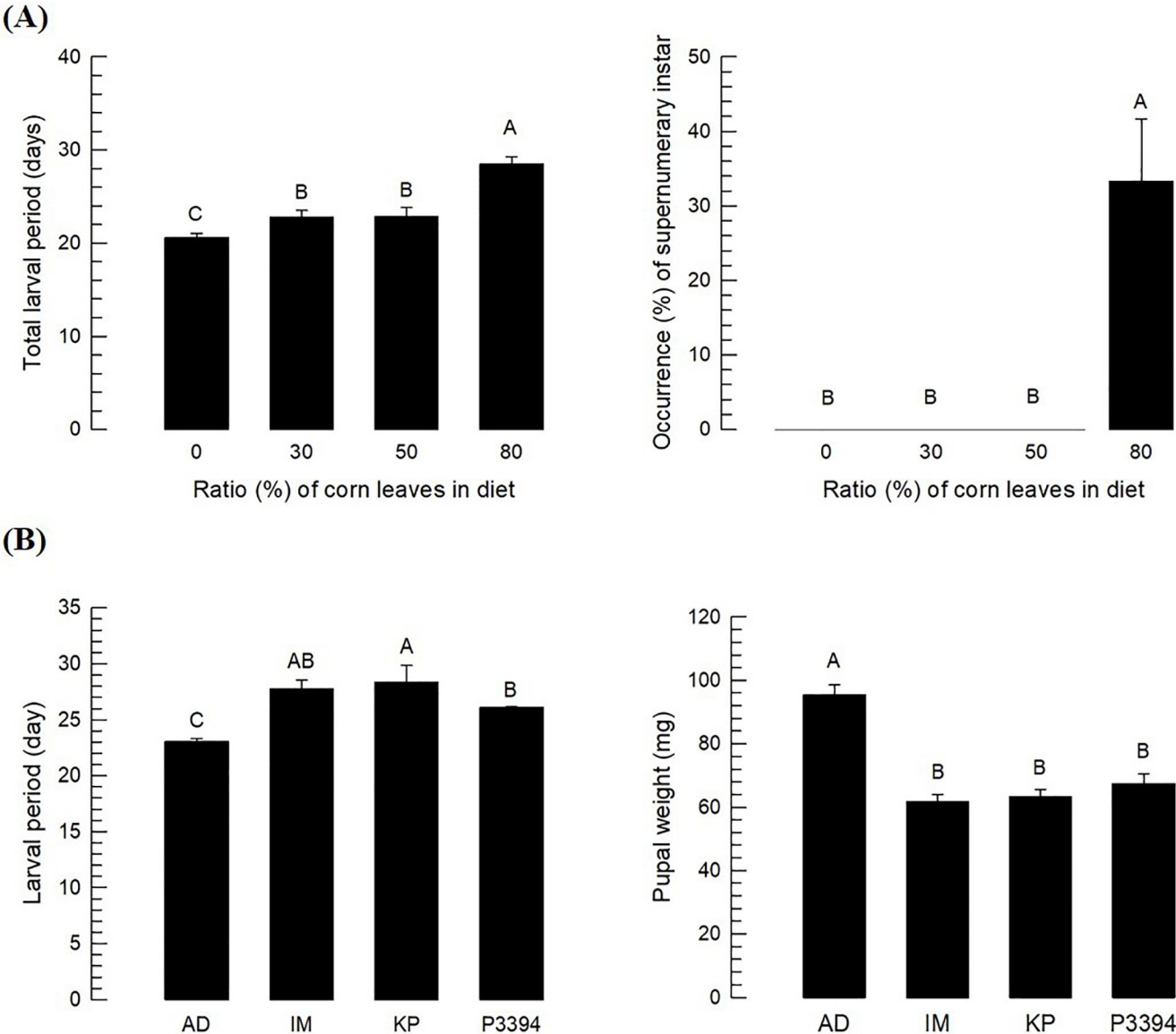

**Fig 2. Influence of resistant corn varieties (Ilmichal ('IM'), Kwangpyeongok ('KP') and P3394) on *O. furnacalis* development by adding to artificial diet.** (A) Effects of KP variety at different doses by adding the freeze-fried leaf powder to artificial diet on larval developmental rate and production of supernumary instar. Newly hatched larvae were reared with different diets. Each treatment was replicated three times. Four larvae were used in each replication. (B) Comparative analysis of three resistant corn varieties on *O. furnacalis* development. Freeze-dried leaf powder of each variety was added to the artificial diet at 80%. Each treatment was replicated three times. Ten larvae were used in each replication. Different letters above standard error bars indicate significant difference among means at Type I error = 0.05 (Tukey test).

resistant corn varieties (Fig 2B). All three resistant varieties resulted in significant ($P < 0.05$) developmental retardation of *O. furnacalis* larvae and reduced their pupal weights.

### 3.3. Three resistant corn varities inhibited chymotrypsin in the gut of *O. furnacalis*

Plant protease inhibitors play crucial roles in antibiosis of insect resistance, especially against digestive chymotrypsin in lepidopteran insects [30, 31]. Chymotrypsin activity in the gut juice of *O. furnacalis* larvae was assessed (Fig 3A). Different protease inhibitors were orally fed to L3

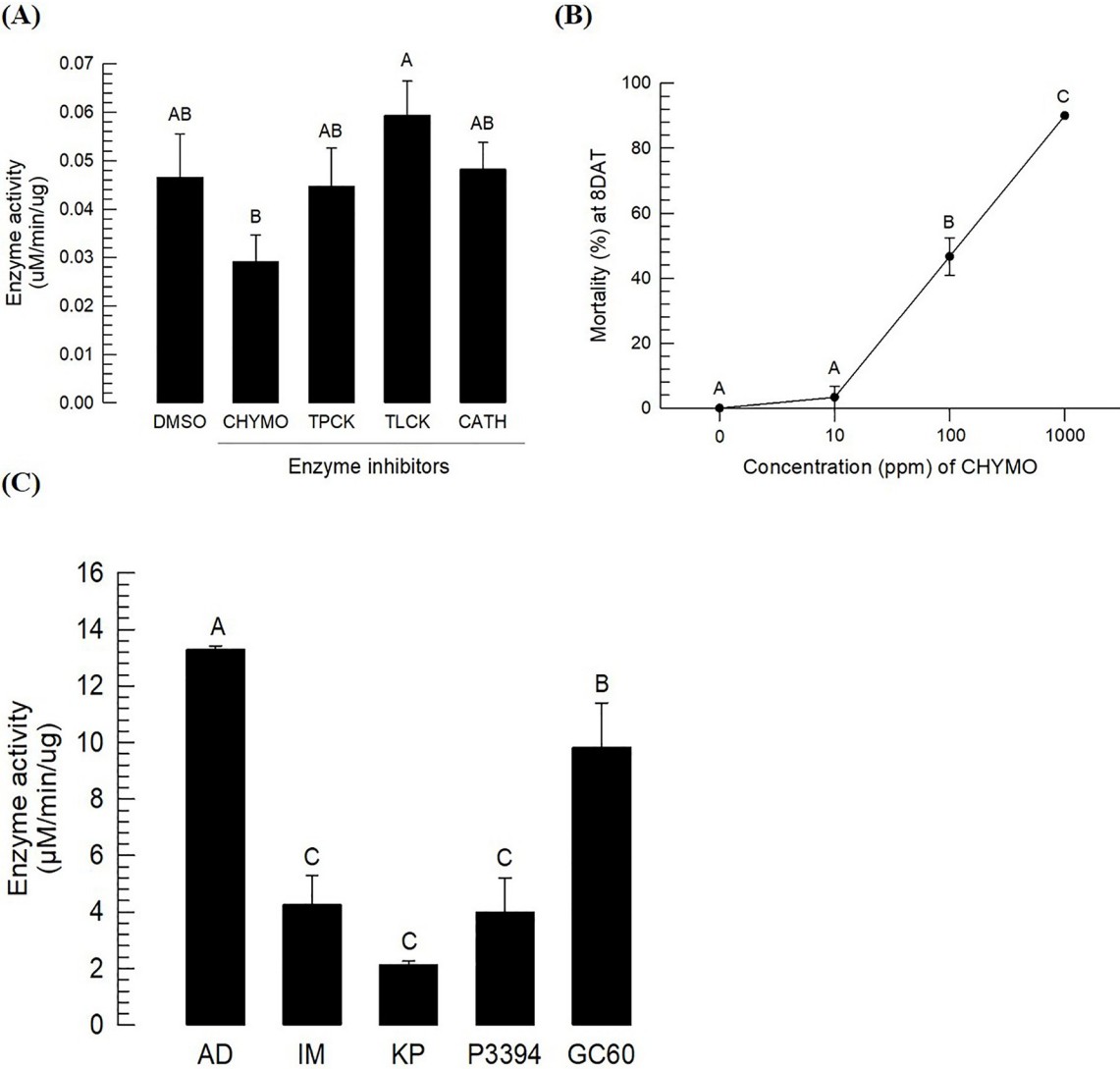

**Fig 3. Inhibitory effects of three resistant corn varieties on chymotrypsin activity in the gut juice of *O. furnacalis* larvae.** (A) Chymotrypsin activity of larvae and sensitivity to its specific inhibitors in L3 larvae. All larvae were fed artificial diet treated with 1,000 ppm inhibitors for 3 days. Three L3 larvae were used to collected gut juice. Each treatment was replicated three times. (B) Survival rate of *O. furnacalis* larvae. L3 larvae were treated with artificial diet treated with different concentrations of inhibitors for 4 days. Survival rates were determined at 8 days after treatment ('8 DAT'). Each treatment was replicated three times. Ten larvae were used in each replication. 'CON' represents solvent (DMSO) treatment without inhibitor. Inhibitors included CHYMO (chymostatin) specific to chymotrypsin; TPCK (tosyl-phenylalanyl chloromethyl ketone) specific to chymotrypsin; TLCK (tosyl-L-lysyl- chloromethane hydrochloride) specific to trypsin, and CATH (cathepsin III inhibitor) specific to cathepsin. (C) Suppression of chymotrypsin activities of L5 larvae fed with leaves of three resistant corn varieties (Ilmichal ('IM'), Kwangpyeongok ('KP'), and P3394). The control used artificial diet ('AD'). Each treatment was replicated three times. Each replication used the midgut juice of three L5 larvae. Different letters above standard error bars indicate significant difference among means at Type I error = 0.05 (Tukey test).

larvae. Four inhibitors (chymostatin and TPCK for chymotrypsin, TLCK for trypsin, and CATH for cathepsin) were used to inhibit different specific proteases. Treatment with chymostatin significantly inhibited the enzyme activity of chymotrypsin. However, the other three inhibitors did not significantly suppress its enzyme activity, although TPCK exhibited a slight but not significant inhibitory activity. The inhibitory activity of chymostatin showed significant lethal effects on L3 larvae of *O. furnacalis* in a dose-dependent manner (Fig 3B). These

results suggest that chymotrypsin enzyme activity is required for the survival of *O. furnacalis* larvae.

The three resistant corn varieties were then assessed for their ability to suppress chymotrypsin activities of *O. furnacalis* (Fig 3C). Compared to larvae reared on artificial diet or the susceptible variety, larvae reared on the three corn varieties suffered, showing much lower chrymotrypsin enzyme activities.

## 3.4. Prediction and expression profile of chymotrypsin genes encoded in the genome of *O. furnacalis*

Nine chymotrypsin genes (*Of-CHY1* ∼ *Of-CHY9*) were predicted from transcriptomes of *O. furnacalis* deposited in GenBank (S3 Table). When these nine *Of-CHY* genes were aligned to construct a phylogenetic tree with other insect CHY genes (Fig 4A), they were separately clustered into three lepidopteran groups. All *CHY*s including *Of-CHY3* were predicted to have conserved domains such as signal peptide, catalytic triad, and substrate binding site (Fig 4B). *Of-CHY*6 and *Of-CHY*7 were identical in their sequences of open reading frame. In this study, six genes (*Of-CHY1*∼*Of-CHY6*) among the candidate genes of *O. furnacalis* were further analyzed for their expression profiles.

These six chymotrypsin genes were expressed in different developmental stages from egg to adults (Fig 5A). *Of-CHY1*, *Of-CHY3*, and *Of-CHY6* genes were highly expressed in the larval stage than in the egg, pupa, or adult stage. This result suggests that these genes may influence larval development. In L5 larvae, all these genes were also expressed in different tissues including midgut, fat body, and epidermis (Fig 5B). Most genes showed relatively higher expression levels in the gut than in the fat body and epidermis except for *Of-CHY1*. Especially, *Of-CHY3* was the most dominant transcript in the midgut. Expression levels of these genes in the midgut of L5 larvae fed different corn leaf were then compared with their levels in the larvae fed with the artificial diet (Fig 5C). In response to feeding on resistant varieties, some CHY genes were induced in their expression while *Of-CHY5* was suppressed in its expression. Especially, larvae fed with a resistant variety (= P3394) up-regulated the expression of *Of-CHY3*.

## 3.5. RNAi of *Of-CHY3* expression led to developmental retardation of *O. furnacalis*

To assess the effect of the major chymotrypsin type (*Of-CHY3*) in the midgut on the development of *O. furnacalis*, RNAi was performed by injecting its specific dsRNA(dsCHY3) to larvae (Fig 6). The dsRNA injection suppressed expression levels of *Of-CHY3* at 24 h post-injection (Fig 6A). This RNAi treatment adversely affected larval survival (Fig 6B), with L4 larvae being highly susceptible to the RNAi treatment, whereas L5 larvae were not.

The higher susceptibility of the 4th instar than 5th instar larvae to the dsRNA treatment allowed us to test younger larvae by the oral dsRNA treatment to avoid a high mechanical damage by injection to young larvae. The oral RNAi efficacy in L3 larvae of *O. furnacalis* was assessed (Fig 7). Oral administration of the dsRNA suppressed the expression level of *OfCHY3* at 12 ∼ 24 h post-feeding compared to that of the control RNAi (Fig 7A). The RNAi treatment was only lethal to L1 larvae (Fig 7B).

## 3.6. Feeding a recombinant *E. coli* expressing dsRNA specific to *Of-CHY3* induces developmental retardation of *O. furnacalis*

A bacterial expression system using *E. coli* HT114 was used to express dsRNA specific to *Of-CHY3* (Fig 8). The dsRNA construct was cloned into L4440 vector (Fig 8A), which expressed

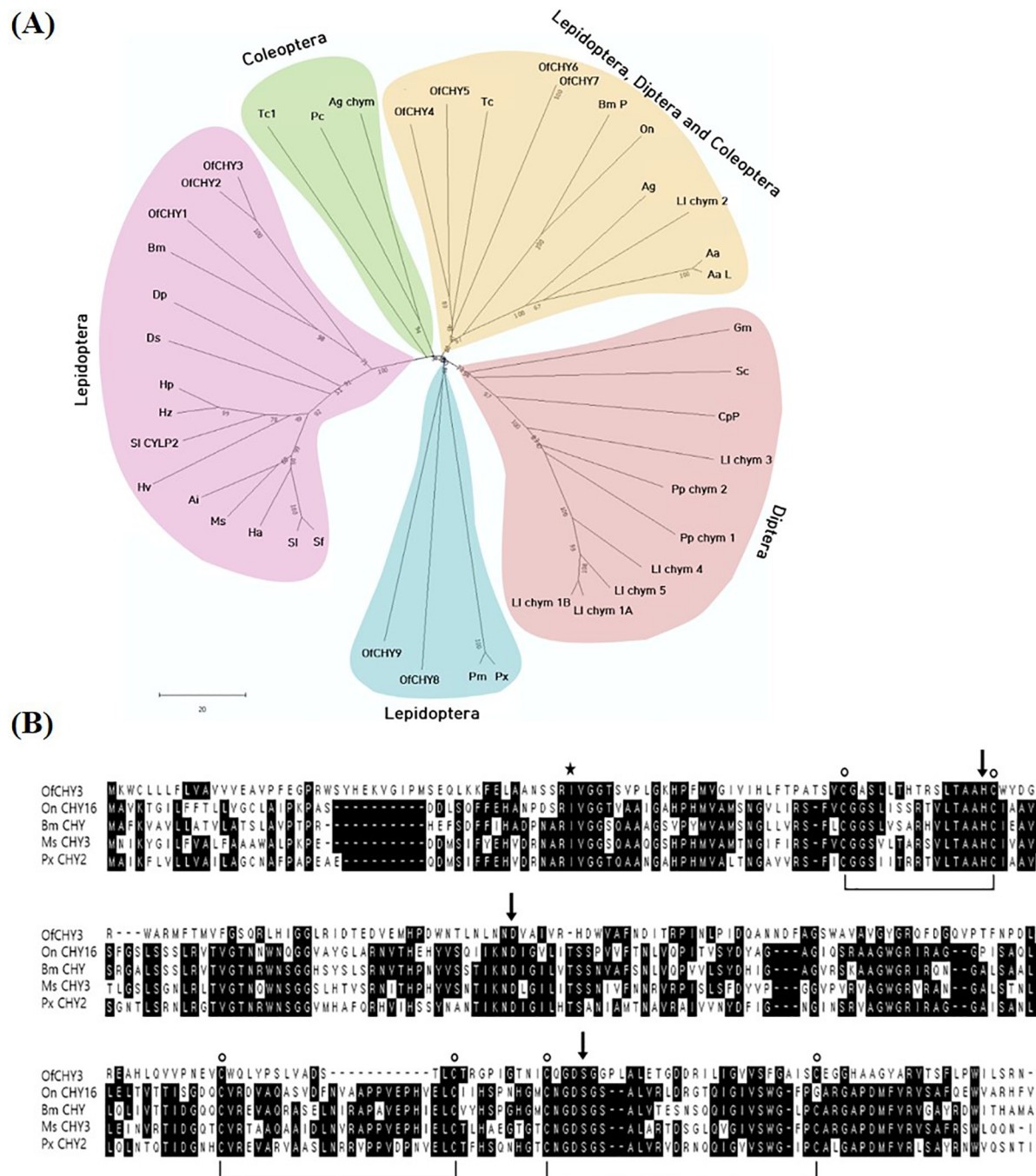

**Fig 4. Molecular characters of chymotrypsins (*Of-CHY1 ~ Of-CHY9*) of *O. furnacalis*.** (A) Phylogenetic analysis with other insect CHYs. Sequence alignment was performed with Clustal W program and the tree was constructed using MEGA9. Each node contains bootstrap value after 1,000 repetitions. GenBank accession numbers are listed in S4 Table. (B) Molecular structure of *Of-CHY3* with other lepidopteran CHY genes. Star indicates cleavage site. Three arrows indicate catalytic triad (H[94], D[141], S[236]). Six dots indicate cysteine residues.

T7 RNA polymerase under lactose promoter and expressed the insert dsRNA construct at the expected size (about 600 bp) (Fig 8B). These recombinant bacteria were fed to teneral larvae of *O. furnalcalis*. The fed larvae exhibited significant reduction of *Of-CHY*3 expression levels (Fig 8C). In all treatments after fed these recombinant bacteria, the expression level of *Of-CHY3*

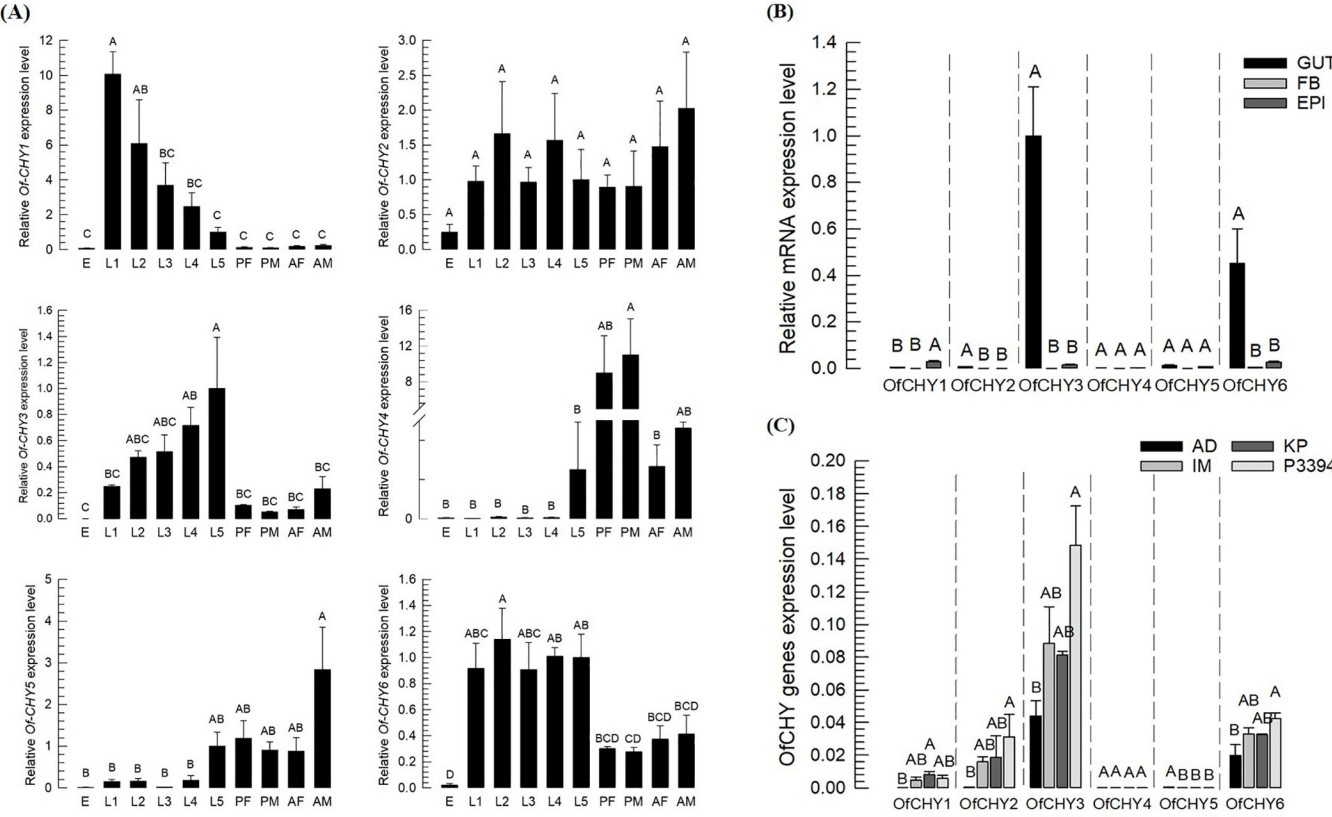

**Fig 5. Expression patterns of six chymotrypsin genes (*Of-CHY1~Of-CHY6*) in *O. furnacalis*.** (A) RT-PCR and RT-qPCR analyses of genes in different developmental stage: Larvae ('L1–L5'), female and male pupae ('PF' and 'PM'), and female and male adults ('AF' and 'AM'). Each treatment was replicated three times. For each replication, 100 eggs, 20 L1 larvae, ten L2 larvae, five L3 larvae, three L4 larvae, one L5 larva, one pupa, and one adult of each sex was used to extracted total RNA. (B) RT-PCR and RT-qPCR of six *Of-CHY* genes in different tissues of *O. furnacalis* reared on artificial diet. Each treatment was replicated three times. For each replication, three L5 larvae was used to extracted total RNA in different tissues: 'GUT' for midgut, 'FB' for fat body, and 'EPI' for epidermis. (C) RT-PCR and RT-qPCR of six *Of-CHY* genes after different diet treatments. Diet treatments included artificial diet ('AD') and three corn varieties including Ilmichal ('IM'), Kwangpyeongok ('KP'), and P3394. Each treatment was replicated three times. For each replication, gut of three L5 larvae reared with different diet was used to extracted total RNA. Different letters above standard error bars indicate significant difference among means at Type I error = 0.05 (Tukey test).

was reduced by about 40% or more. Compared to larvae reared on non-reombinant diet, larvae fed recombinant bacteria had significantly lower CHY enzyme activities (Fig 8D). Insecticial activities of these recombinant bacteria were assessed against different larval instars. Insecticidal activity against young larval stage was found. At a dose of $10^8$ cells per larva, over 70% of mortality was observed at L1 stage (Fig 8E).

## 4. Discussion

Obtaining insect resistance is an optimal strategy to control target herbivorous insects in agriculture [32]. Antixenosis and antibiosis are two main mechanisms involved in insect resistance. Antixenosis is defined as nonpreference of host plants against infestation of target insects [33]. In contrast, antibiosis is induced by toxic compound(s) to give adverse effects on target insects [4, 5]. Tolerance is another mechanism to express insect resistance by excessive production of specific plant parts, which would be unnecessary for a normal plant growth and reproduction [6]. Our current study evaluated insect resistance in different corn varieties against *O. furnacalis*.

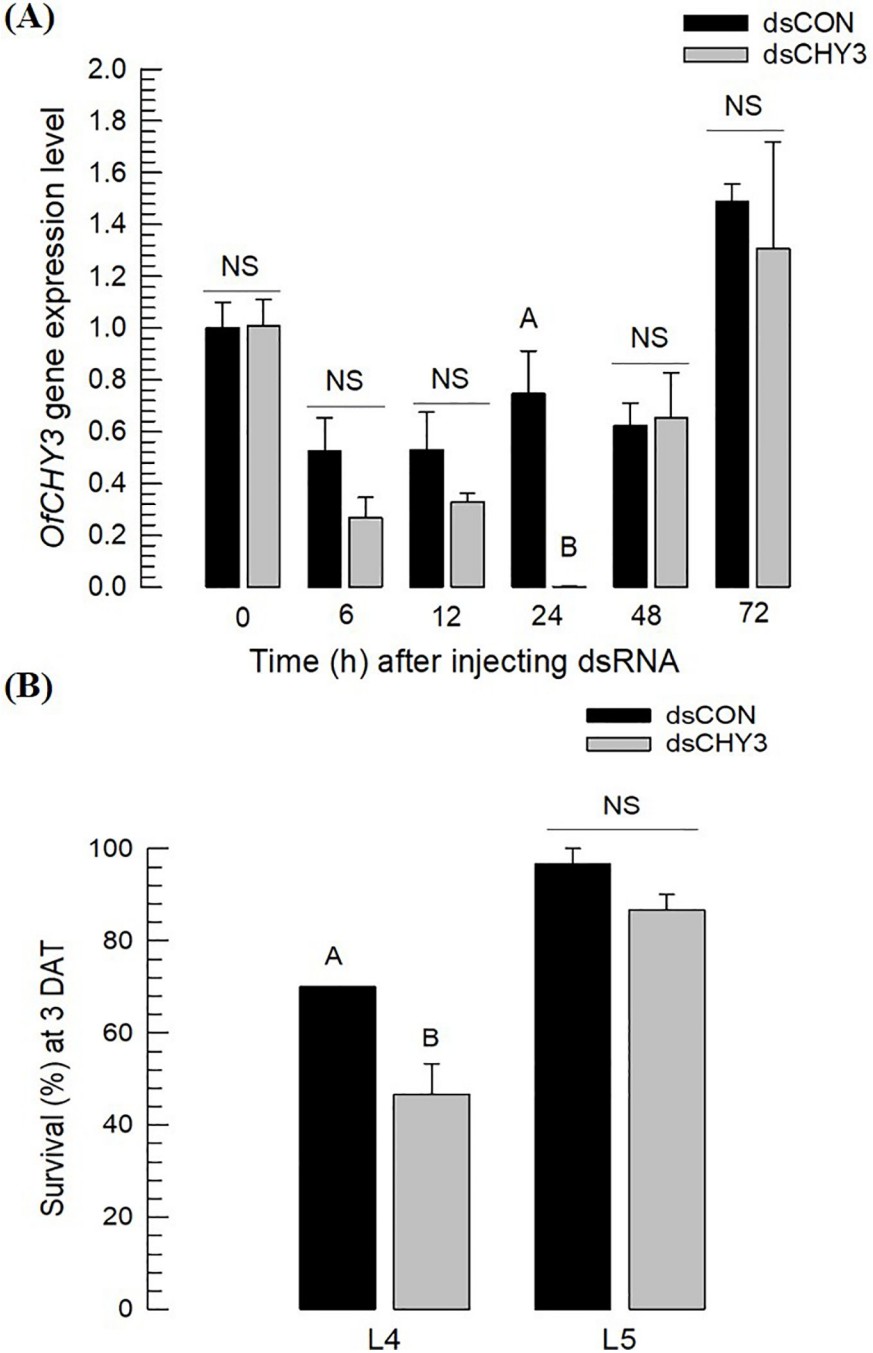

**Fig 6. RNAi of *Of-CHY3* by injecting its dsRNA to L5 larvae of *O. furnacalis*.** Each larva was injected 2 μg of dsRNA. As control, the same amount of dsRNA ('dsCON') specific to EGFP was injected. (A) Change in expression levels of *OfCHY3* after dsRNA injection to L5 larvae. RT-qPCR used RNA samples collected from whole body. (B) Effect of dsRNA injection on survival of *O. furnacalis* larvae at different stages. Survival rate was determined at three days after treatment ('3 DAT'). Each treatment was replicated three times. Ten larvae were used in each replication. Different letters above standard error bars indicate significant difference among means at Type I error = 0.05 (Tukey test).

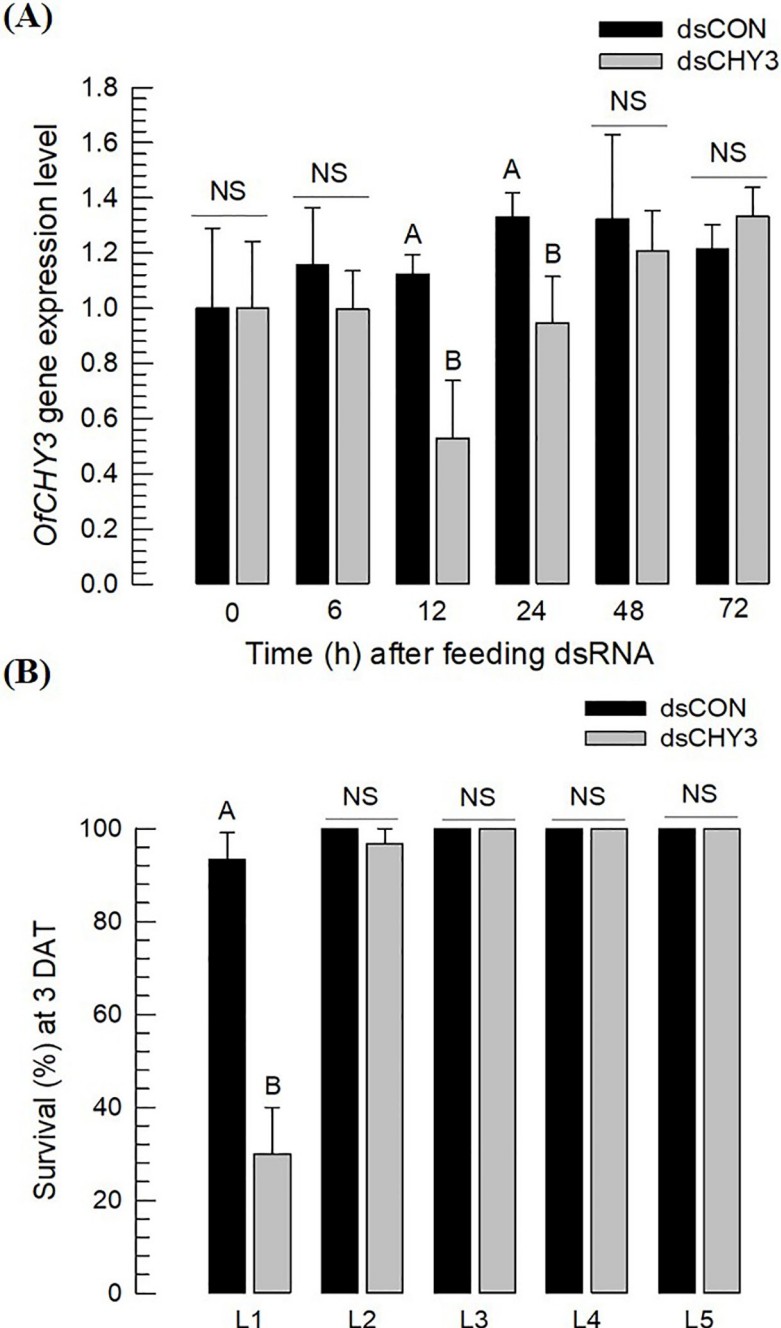

**Fig 7. RNAi of *Of-CHY3* by feeding its dsRNA to L3 larvae of *O. furnacalis*.** Each larva was fed 10 μg of dsRNA. As control, the same amount of dsRNA ('dsCON') specific to EGFP was fed. (A) Changes in expression levels of *OfCHY3* after oral administration of dsRNA injection to L3 larvae. RT-qPCR used RNA samples collected from whole body. (B) Effects of feeding dsRNA on survival of *O. furnacalis* larvae at different stages. Survival rate was determined at three days after treatment ('3 DAT'). Each treatment was replicated three times. Ten larvae were used in each replication. Different letters above standard error bars indicate significant difference among means at Type I error = 0.05 (Tukey test).

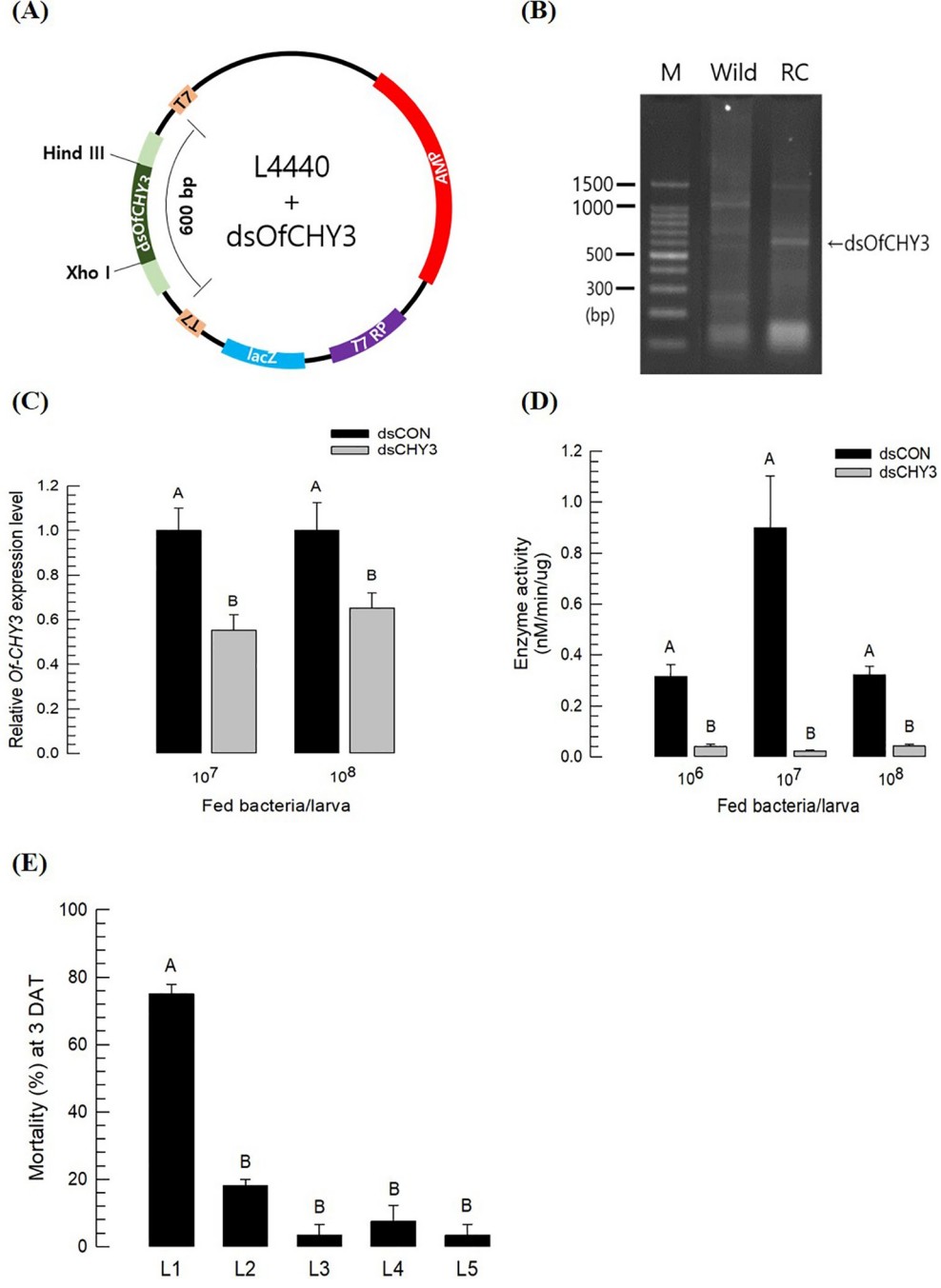

**Fig 8. Influence of recombinant *E. coli* expressing dsRNA specific to *Of-CHY3* ('dsOfCHY3').** (A) Cloning map of recombinant vector (L4440-dsOfCHY3). (B) Expression of dsOfCHY3 by bacteria. 'Wild' and 'RC' represent non-recombinant (L4440) and recombinant (L4440-dsOfCHY3) HT115 bacteria, respectively. 'M' represents DNA size marker. (C) Suppression in expression level of *Of-CHY3* after recombinant bacterial feeding. (D) Suppression of chymotrypsin activity in the gut juice of *O. furnacalis* larvae fed with recombinant bacteria. Each treatment was replicated three times. For each treatment, three L4 larvae were used to extract total RNA and collect gut juice. (E) Mortality of *O. furnacalis* larvae at different larval stages. All treatments were used for oral administration of the recombinant bacteria by loading $10^8$ cells to the artificial diet. Mortality was determined at 3 days after treatment ('3 DAT'). Each treatment was replicated three times. For each replication, ten larvae were used. Developmental alteration after bacterial feeding was determined. Different letters above standard error bars indicate significant difference among means at Type I error = 0.05 (Tukey test).

Three resistant corn varieties were selected after screening damage by *O. furnacalis*. Compared to the susceptible variety ('GC60'), three resistant varieties were not significantly different in host preference of *O. furnacalis*. However, compared to GC60, these three corn varieties significantly altered larval development such as developmental retardation, induction of supernumary larval instars, and high larval mortality. High larval mortality and induction of supernumary larval instars might be caused by nutritional deficiency due to poor digestion, which could cause not enough larval growth to reach a certain critical body size for larva-to-pupal metamorphosis [34]. These results suggest that their insect resistance comes from antibiosis, but from antixenosis. The antibiosis hypothesis was further supported by additional experiments using corn leaves to treat artificial diet. Artificial diet was optimal for larval growth of *O. furnacalis* [21]. The addition of resistant corn leaves to the artificial diet prevented larval development in a dose-dependent manner. These results support the antibiosis of these corn varieties involved in their insect resistance against *O. furnacalis*.

There were significant decreases in chymotrypsin enzyme activity in gut lumen of *O. furnacalis* after feeding resistant corn varieties. In lepidopteran insects, protein digestion mostly depends on catalytic activity of serine proteases along with catalytic activities of cysteine proteases, carboxypeptidases, and aminopeptidases [18]. These serine proteases include trypsin and chymotrypsin identified in midgut transcriptomes of lepidopterans [35]. Among 120 serine proteases annotated in *Plutella xylostella*, for example, 38 trypsins and 8 chymotrypsins have been reported [36]. In fact, RNAi of these serine proteases resulted in significant mortalities along with reduced body size, supporting their physiological roles in digestion [31]. These results suggest that the decrease of chymotrypsin activity in *O. furnacalis* larvae after feeding resistant corn varieties can explain the reduced larval development. Furthermore, these resistant corn varieties may produce serine protease(s) inhibitory factor(s) to suppress the activity of chymotrypsin. In addition to this constitutive resistant factor(s), these corn varieties may induce other resistant factors. For example, *O. furnacalis*-induced corns can adversely affect subsequent feeding of larvae by causing retardation of immature development and suppressing fecundity of female adults [37]. Subsequent biochemical analysis has indicated an increase in 2-(2-hydroxy-4,7-dimethoxy-1,4-benzoxazin-3-one)-β-D-glucopyranose (HDIMBOA-Glc) along with higher activities of defensive enzymes such as peroxidase, superoxide dismutase, catalase, and polyphenol oxidase [37].

Nine CHY genes (*Of-CHYs*) were predicted from *O. furnacalis*, in which six genes were confirmed in their expression in different developmental stages and tissues. *Of-CHY3* was highly expressed in the midgut of *O. furnacalis* larvae. Its RNAi using dsRNA specific to *OfCHY3* showed significant adverse effects on larval survival, especially on young larvae of *O. furnacalis*. This lethal effect was also observed when the dsRNA was fed to larvae. This suggests that the enzyme activity is required for the development of *O. furnacalis* because Of-CHY3 is likely to be the main source of the enzyme activity in the midgut. This is further supported by the larval lethalilty activity after feeding recombinant *E. coli* overexpressing dsRNA specific to *Of-CHY3*.

Altogether, these results suggest that the three varieties of corns are resistant to *O. furnacalis* by exhibiting antibiosis using chymotrypsin-inhibitory activities. The antibiosis by inhibiting the enzyme activity was demonstrated by a loss-of-functional study using RNAi. However, the compound(s) that inhibited the enzyme activity was(were) not identified from these resistant corn varieties in this study. Among secondary metabolites produced by resistant corns, cyclic hydroxamic acids (cHx), which are plant-derived benzoxazinoids (BXs), have been proposed as the antibiosis factor(s) because they are produced by a number of gramineous plants including corn, wheat, and rye. They are toxic to many herbivores [9, 38]. cHx are usually stored in plants as non-toxic glucosides after glycosylation [39, 40]. However, upon damage, they are

activated into toxic aglucon by glycosidases [41]. DIMBOA, the main cHx in corn, has been associated with resistance to herbivorous insects including *O. furnacalis* [42]. DIMBOA can act as a feeding deterrent. It is also a toxin to various insects by inhibiting digestive proteases including chymotrypsin [43]. Thus, the three resistant corn varieties determined in this study might produce a large amount of cHx metabolite(s) to inhibit chymotrypsin activity of *O. furnacalis* and inhibit their larval growth.

## Supporting information

**S1 Table. Artificial diet (1 L) treatments containing three different corn varieties: Ilmichal (IM), Kwangpyeongok (KP), and P3394.**
(DOCX)

**S2 Table. List of primers used in this study.**
(DOCX)

**S3 Table. Putative chymotrypsin (CHY) genes of *O. furnacalis*.**
(DOCX)

**S4 Table. GenBank accession numbers and abbreviations used for phylogenetic analysis.**
(DOCX)

## Acknowledgments

We would like to thank Duyeol Choi for his technical support and constructive advices on statistical analyses.

## Author Contributions

**Conceptualization:** Yonggyun Kim.

**Data curation:** Jin Kyo Jung.

**Formal analysis:** Jin Kyo Jung.

**Funding acquisition:** Eun Young Kim, Yonggyun Kim.

**Investigation:** Eun Young Kim, Jin Kyo Jung, I. Hyeon Kim, Yonggyun Kim.

**Methodology:** Eun Young Kim, Jin Kyo Jung, I. Hyeon Kim.

**Resources:** Eun Young Kim, I. Hyeon Kim.

**Software:** Eun Young Kim.

**Supervision:** Jin Kyo Jung, Yonggyun Kim.

**Validation:** Eun Young Kim.

**Visualization:** Eun Young Kim.

**Writing – original draft:** Eun Young Kim, Yonggyun Kim.

**Writing – review & editing:** Yonggyun Kim.

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
