## [Decision Letter · Decision Letter 0]

9 Mar 2022

PONE-D-22-02777Chymotrypsin is a Molecular Target of Insect Resistance of Three Corn Varieties Against the Asian Corn Borer, Ostrinia furnacalisPLOS ONE

Dear Dr. Kim,

Thank you for submitting your manuscript to PLOS ONE. After careful consideration, we feel that it has merit but does not fully meet PLOS ONE’s publication criteria as it currently stands. Therefore, we invite you to submit a revised version of the manuscript that addresses the points raised during the review process.

Please improve the manuscript following the comments of the two reviewers. ==============================

We look forward to receiving your revised manuscript.

Kind regards,

Yonggen Lou

Academic Editor

PLOS ONE

Journal Requirements:

 [This research was supported by a “Cooperatibe Research Program for Agriculture Science & Technology Development (Project No. PJ01503802)” funded by Rural Development Administration, Republic of Korea to EYK. This work was also supported by a Research Grant of Andong National University, Republic of Korea to YK.]

Reviewers' comments:

Reviewer's Responses to Questions

**Comments to the Author**

1. Is the manuscript technically sound, and do the data support the conclusions?

Reviewer #1: Yes

Reviewer #2: Yes

2. Has the statistical analysis been performed appropriately and rigorously? 

Reviewer #1: Yes

Reviewer #2: Yes

3. Have the authors made all data underlying the findings in their manuscript fully available?

Reviewer #1: Yes

Reviewer #2: Yes

4. Is the manuscript presented in an intelligible fashion and written in standard English?

Reviewer #1: Yes

Reviewer #2: Yes

5. Review Comments to the Author

Reviewer #1: In this manuscript, the authors selected three corn varieties (‘IM’, ‘KP’, and P3394) with resistance to Ostrinia furnacalis from 27 corn varieties. Further insect feeding preference test and growth parameter determination revealed that these three resistant varieties influenced O. furnacalis development rather than food preference. Further, the authors compared the response of O. furnacalis after ingestion of these three insect-resistant corn varieties, susceptible variety CG60, or artificial diet. O. furnacalis on insect-resistant corn varieties exhibited low chymotrypsin (CHY) activity and reduced CHY gene expression, especially Of-CHY3. RNA interference (RNAi) of Of-CHY3 by dsRNA injection or feeding decreased Of-CHY3 expression and suppressed CHY activity. In addition, the authors also checked the expression patterns of Of-CHY1 to Of-CHY6 in different developmental stage of O. furnacalis. Finally, the authors fed O. furnacalis with the recombinant Escherichia coli expressing dsRNA specific to Of-CHY3, and larval development retardation of O. furnacalis was observed. The results in this study provided new target in O. furnacalis for breeding insect-resistant corns. The manuscript is interesting but some major points that should be clarified by authors.

1. O. furnacalis mainly feed on corn stems, but in this study, the authors used corn leaves as food for a series of analysis, do corn stems and leaves contain same chymotrypsin inhibitory compounds?

2. For the RNAi of Of-CHY3, the dsRNA injection was performed on L5 of O. furnacalis, while the dsRNA feeding experiment was done on L3 of O. furnacalis. Why were these experiments done on different stages of O. furnacalis?

3. In Fig. 5B, it is better to use the full forms of FB and EPI in corresponding figure legends.

4. In Table 1, please indicate what the rows in light green represent in the footnotes.

Reviewer #2: Review of the manuscript: Chymotrypsin is a Molecular Target of Insect Resistance of Three Corn Varieties Against the Asian Corn Borer, Ostrinia furnacalis

My overall comments are as follow:

1. The results are overall clearly presented.

2. Some detailed information in the experiment should be provided.

3. Mistakes in text and figure should be fixed.

Line 68-69, maybe it is better to change a another example. This citation is more focus on systemic defense induced by root feeder than defensive metabolism induced by leaf herbivore such as corn borers.

Line 73, “Defense-related enzymes in plants” may be more specific.

Line 104, the corn variety GC60 should be attached here.

Line 154, “Feeding amount of each leaf was measured at 2 days” Please be more clear about how the measurement was done and which picture analyzing software was used.

Line 155, How many petri dishes were used in each time of the feeding test? Ten larvae per petri dish and each treatment was replicated three times, so in total 30 larvae was tested?

Line 165, “freeze-dried” by using what? Corn leaves were ground in liquid N2? Has the AD been boiled for sterilization? Has the corn leaf powder been added to the AD during the cooling step of the AD preparation? Detailed information should be provided.

Line 169-170, same question as above. How these proteases had been added to the AD, during the cooling step of the AD?

Line 198-199, How these CHY-like genes were identified? By using blastX? Has any template gene (such as a identified CHY-like gene in some model organism such as fruit fly) been used in the blast?

Line 212, “For each type of tissues, three larvae were used” the sample size is a little small.

Line 256, “treated with dsOfCHY3 solution” How the treatment was done? A drop of dsOfCHY3 solution was added to the AD? Or, the dsRNA were diluted into distilled water to make a solution and then the AD was soaked by this solution?

Line 361, “is required for the development” maybe “survival” is more precise here instead of the word ”development”

Line 372-373, “All Of-CHYs were predicted to have…” in fig. 4B there is only Of-CHY3.

Line 378, “All genes showed relatively higher expression levels in L5 stage” this statement seems not correct based on the expression pattern of these genes.

Line 387, “larvae fed IM” Based on the color of the bars, “IM” should be replaced by “P3394”?

Line 393-394, the knockdown efficiency of dsRNA injection was analyzed based on the gene expression level in the full body or gut of the insect?

Table 1, dash means no record?

Line 639, 80% in diet, I guess this represent 80% of 32.2 g corn leaf powder in 1 L AD?

Line 655-656, the suppression of chymotrypsin activities were tested by using larvae fed on corn leaves or AD with corn leaf powder?

6. PLOS authors have the option to publish the peer review history of their article (what does this mean?). If published, this will include your full peer review and any attached files.

Reviewer #1: No

Reviewer #2: No

---

## [Author Response · Author response to Decision Letter 0]

10 Mar 2022

Response to reviewers’ comments

[Reviewer #1]

Comment #1-1: O. furnacalis mainly feed on corn stems, but in this study, the authors used corn leaves as food for a series of analysis, do corn stems and leaves contain same chymotrypsin inhibitory compounds?

Response: It is a very nice comment. O. furnacalis larvae usually feed on stems and prefer stems to leaves. We presumed that antibiosis factor(s) is localized in both stem and leaves. Based on this assumption, we screened the corn varieties by treating leaves of all test varieties. Especially, this study used leaves near to stem to stimulate feeding activity. This information is added to the M&M text as follows: “Leaves of four corn varieties were cut at the proximal area containing the stalk, which was wrapped with wet cotton to prevent desiccation.” 

Comment #1-2: For the RNAi of Of-CHY3, the dsRNA injection was performed on L5 of O. furnacalis, while the dsRNA feeding experiment was done on L3 of O. furnacalis. Why were these experiments done on different stages of O. furnacalis?

Response: To confirm RNAi efficiency, this study used dsRNA injection and feeding methods. Results showed that both dsRNA delivery methods were efficient to suppress target gene, Of-CHY3. To test insecticidal activities of the RNAi treatments, we compared 4th instar and 5th instar larvae in susceptibility to the RNAi injection and showed the younger larvae were more susceptible. Based on this observation, we applied the dsRNA to younger larvae from 1st instar larvae. This information is added to the Result as follows: “The higher susceptibility of the 4th instar than 5th instar larvae to the dsRNA treatment allowed us to test younger larvae by the oral dsRNA treatment to avoid a high mechanical damage by injection to young larvae.” 

Comment #1-3: In Fig. 5B, it is better to use the full forms of FB and EPI in corresponding figure legends.

Response: The acronyms are explained in the figure caption as follows: “For each replication, three L5 larvae was used to extracted total RNA in different tissues: ‘GUT’ for midgut, ‘FB’ for fat body, and ‘EPI’ for epidermis.”

Comment #1-4: In Table 1, please indicate what the rows in light green represent in the footnotes.

Response: The information is added as follows: “Red- and blue-colored varieties were selected as resistant and susceptible varieties for subsequent study.” 

 

[Reviewer #2]

Comment #2-1: Line 68-69, maybe it is better to change a another example. This citation is more focus on systemic defense induced by root feeder than defensive metabolism induced by leaf herbivore such as corn borers.

Response: It is replaced as follows: “For example, 2,4-dihydroxy-7-methoxy-1,4-benzoxazine-3-one, commonly known as DIMBOA, has been reported in several gramineous species, including maize, wheat, and rye [9].” 

Comment #2-2: Line 73, “Defense-related enzymes in plants” may be more specific.

Response: Corrected as suggested

Comment #2-3: Line 104, the corn variety GC60 should be attached here.

Response: Added 

Comment #2-4: Line 154, “Feeding amount of each leaf was measured at 2 days” Please be more clear about how the measurement was done and which picture analyzing software was used.

Response: It was measured by measuring weight loss. This information is added to the M&M as follows: “Feeding amount of each leaf was measured by weight loss for 2 days after initiation of the trial. The feeding amount was corrected by the weight loss of the same size of leaf due to desiccation under the same environmental conditions during the assay.” 

Comment #2-5: Line 155, How many petri dishes were used in each time of the feeding test? Ten larvae per petri dish and each treatment was replicated three times, so in total 30 larvae was tested?

Response: The information was written, but modified as follows: “In a treatment (= Petri dish), 10 starved larvae were placed in the center of the dish. Each treatment was replicated three times.” 

Comment #2-6: Line 165, “freeze-dried” by using what? Corn leaves were ground in liquid N2? Has the AD been boiled for sterilization? Has the corn leaf powder been added to the AD during the cooling step of the AD preparation? Detailed information should be provided.

Response: The detailed information is added as follows: “corn leaves were freeze-dried by Biobase (FD8508, Ilshin, Dongducheon, Korea) and pulverized. The free-dried powder was then added to the artificial diet (Table S1).” 

Comment #2-7: Line 169-170, same question as above. How these proteases had been added to the AD, during the cooling step of the AD?

Response: The information is modified as follows: “All inhibitors were dissolved in 10% dimethyl sulfoxide (DMSO) to prepare stock solution at 10, 100, and 1,000 ppm. L3 larvae were fed artificial diet (0.2 x 2.0 x 0.1 cm) overlaid with 30 uL inhibitor solution for 4 days.” 

Comment #2-8: Line 198-199, How these CHY-like genes were identified? By using blastX? Has any template gene (such as a identified CHY-like gene in some model organism such as fruit fly) been used in the blast?

Response: They were already annotated. Thus the word was changed into “obtained”’

Comment #2-9: Line 212, “For each type of tissues, three larvae were used” the sample size is a little small.

Response: Due to relatively big body size of fifth instar larvae as mentioned above, three larvae were enough to collect tissues. 

Comment #2-10: Line 256, “treated with dsOfCHY3 solution” How the treatment was done? A drop of dsOfCHY3 solution was added to the AD? Or, the dsRNA were diluted into distilled water to make a solution and then the AD was soaked by this solution?

Response: To be clear, the sentence is rephrased as follows: “For dsRNA oral delivery, 6 h-starved L3 larvae were treated with an artificial diet (0.2 x 2.0 x 0.1 cm) overlaid with dsOfCHY3 solution. After a complete dsRNA-treated diet consumption (within 24 h), the consumed dsRNA concentration was calculated by dividing by the number of treated larvae per diet.” 

Comment #2-11: Line 361, “is required for the development” maybe “survival” is more precise here instead of the word ”development”

Response: It is a nice suggestion. It is replaced with survival.

Comment #2-12: Line 372-373, “All Of-CHYs were predicted to have…” in fig. 4B there is only Of-CHY3.

Response: Rephrased as follows: “All CHYs including Of-CHY3” 

Comment #2-13: Line 378, “All genes showed relatively higher expression levels in L5 stage” this statement seems not correct based on the expression pattern of these genes.

Response: The sentence is deleted. 

Comment #2-14: Line 387, “larvae fed IM” Based on the color of the bars, “IM” should be replaced by “P3394”?

Response: It is changed into P3394 as follows: “Especially, larvae fed with a resistant variety (= P3394) up-regulated the expression of Of-CHY3.”. 

Comment #2-15: Line 393-394, the knockdown efficiency of dsRNA injection was analyzed based on the gene expression level in the full body or gut of the insect?

Response: It was from a whole body. We add the information to M&M as follows: “RT-qPCR used RNA samples collected from whole body.” 

Comment #2-16: Table 1, dash means no record?

Response: We added the information as follows: “Red- and blue-colored varieties were selected as resistant and susceptible varieties for subsequent study.” 

Comment #2-17: Line 639, 80% in diet, I guess this represent 80% of 32.2 g corn leaf powder in 1 L AD?

Response: Rephrased as follows: “Comparative analysis of three resistant corn varieties on O. furnacalis development. Freeze-dried leaf powder of each variety was added to the artificial diet at 80%.” 

Comment #2-18: Line 655-656, the suppression of chymotrypsin activities were tested by using larvae fed on corn leaves or AD with corn leaf powder?

Response: Clarified as follows: “Suppression of chymotrypsin activities of L5 larvae fed with leaves of three resistant corn varieties”

---

## [Editor Report · Decision Letter 1]

28 Mar 2022

Chymotrypsin is a Molecular Target of Insect Resistance of Three Corn Varieties against the Asian Corn Borer, Ostrinia furnacalis

PONE-D-22-02777R1

Dear Dr. Kim,

We’re pleased to inform you that your manuscript has been judged scientifically suitable for publication and will be formally accepted for publication once it meets all outstanding technical requirements.

Kind regards,

Yonggen Lou

Academic Editor

PLOS ONE
---

## [Editor Report · Acceptance letter]

30 Mar 2022

PONE-D-22-02777R1 

Chymotrypsin is a Molecular Target of Insect Resistance of Three Corn Varieties against the Asian Corn Borer, *Ostrinia furnacalis*

Dear Dr. Kim:

I'm pleased to inform you that your manuscript has been deemed suitable for publication in PLOS ONE. Congratulations! Your manuscript is now with our production department. 

Kind regards, 

on behalf of

Dr. Yonggen Lou 

Academic Editor

PLOS ONE